# Mental healthcare utilisation by patients before and after receiving paliperidone palmitate treatment: mirror image analyses

Giouliana Kadra-Scalzo [1], Deborah Ahn,[1] Alex Bird,[2] Matthew Broadbent,[3] Chin-Kuo Chang,[1,4] Megan Pritchard,[3] Hitesh Shetty,[3] David Taylor,[3] Richard Hayes,[1] Robert Stewart[1,3]

RH and RS contributed equally.

[1]Psychological Medicine, King's College London, London, UK
[2]Janssen Pharmaceutical Companies of Johnson & Johnson, Titusville, New Jersey, USA
[3]South London and Maudsley NHS Foundation Trust. London, UK, London, UK
[4]Institute of Epidemiology and Preventive Medicine, College of Public Health, National Taiwan University, Taipei, Taiwan

**Correspondence to**
Dr Giouliana Kadra-Scalzo; giouliana.kadra@kcl.ac.uk

## ABSTRACT

**Objectives** To compare mental healthcare use and healthcare professional (HCP) contacts for patients before and after initiation of paliperidone palmitate.

**Setting** The South London and Maudsley NHS Foundation Trust (SLAM) Biomedical Research Centre Clinical Record Interactive Search.

**Participants** We identified all adults with a diagnosis of schizophrenia (International Classification of Diseases 10th Revision: F20.x), who had received paliperidone palmitate prescription for at least 365 days and had at least 1 year of recorded treatment from SLAM, prior to the first recorded receipt of paliperidone palmitate.

**Primary and secondary outcome measures** Inpatient and community mental healthcare service use, such as inpatient bed days, number of active days in the service, face-to-face and telephone HCP use in the 12 months before and after paliperidone palmitate initiation.

**Results** We identified 664 patients initiated on paliperidone palmitate. Following initiation, inpatient bed days were lower, although patients remained active on the service case load longer for both mirror approach 1 (mean difference of inpatient bed days −10.48 (95% CI −15.75 to −5.22); days active 40.67 (95% CI 33.39 to 47.95)) and mirror approach 2 (mean difference of inpatient bed days −23.96 (95% CI −30.01 to −17.92); mean difference of days active 40.69 (95% CI 33.39 to 47.94)). The postinitiation period was further characterised by fewer face-to-face and telephone contacts with medical and social work HCPs, and an increased contact with clinical psychologists.

**Conclusions** Our findings indicate a change in the profile of HCP use, consistent with a transition from treatment to possible rehabilitation.

## BACKGROUND

Treatment discontinuation and covert nonadherence is high among patients with serious mental illnesses[1] and current guidelines for schizophrenia recommend that depot medication can be considered[2] to reduce relapse. For patients who have been previously non-adherent, depot antipsychotics have been

### Strengths and limitations of this study

► We captured a large and diverse sample of patients in naturalistic settings, reflective of real-life clinical practice.
► We employed two methods of mirror image analyses to address the potentially biasing impact of the in-patient episodes within which many initiation events take place.
► The study was observational in nature and therefore the before-after comparisons cannot be assumed to represent causal relationships.
► We were unable to examine the effects of factors, which were not captured by the routine health records.

associated with lower risk of rehospitalisation[3 4] and relapse[5] in comparison with oral preparations of the same medications and placebos.[6] However, there have been significant concerns over extrapyramidal effects[7–9]; and the introduction of second-generation (atypical) depot antipsychotics was cited as a potentially significant opportunity for a lower risk of movement disorders.[10]

Research into second-generation depot antipsychotics has been relatively sparse. Findings from studies examining risperidone long-acting injection have been mixed with some evidence indicating that risperidone depot is associated with reduced hospital admissions, total inpatient days[11] and improvement in clinical symptoms,[12] whereas other studies have suggested that risperidone depot either made no difference[13] or was associated with increased bed stay and healthcare costs.[14]

Paliperidone palmitate was introduced in the UK National Health Service in 2011 and is a second-generation antipsychotic depot typically administered once a month intramuscularly. Several studies have

investigated the clinical efficacy of paliperidone palmitate in treating schizophrenia symptoms.[15 16] Although paliperidone palmitate is associated with higher acquisition cost in comparison to other oral antipsychotics,[17] several observational studies have investigated its effectiveness and reported favourable findings in relation to inpatient hospitalisation. For example, Bressington and colleagues[18] reported a reduced number of acute inpatient admissions a year after paliperidone palmitate initiation. Similarly, Taylor and Olofinjana[3] reported a significant decrease in the number and length of hospital admissions per patient per year. Therefore, it appears that although paliperidone palmitate may be associated with a higher initial cost, this may be offset over time through a decrease in rehospitalisation. However, although hospitalisation episodes are strong determinants of healthcare costs,[14] it is also important to understand healthcare professional (HCP) contacts, which reflect intensity of input required in outpatient settings. To the authors' knowledge, there has been no research to date that has examined secondary mental healthcare use in this level of detail. Consequently, the aim of this study was to describe inpatient and community mental healthcare service use, using observational data including HCP contacts, of patients with schizophrenia being treated with paliperidone palmitate, specifically comparing the frequency and duration of HCP contacts before and after initiation of paliperidone palmitate therapy, using mirror image methodologies. On one hand, observational data are limited with respect to trial data because of the lack of randomisation and thus the risk of residual confounding; on the other hand, they provide an opportunity to assess associations in a more naturalistic and generalisable setting than is generally the case for trials.

## METHODS

The data for this study were sourced from the South London and Maudsley NHS Foundation Trust (SLAM) Biomedical Research Centre Clinical Record Interactive Search (CRIS). CRIS has been previously described in detail[19 20]; briefly, this is a data resource sourced from the electronic mental health records of SLAM, which provides comprehensive mental health services to a geographic catchment of 1.36 million residents in four boroughs of south London. All records are deidentified by CRIS and made available for research use under the governance framework.[21] CRIS data have been substantially enhanced through the application of natural language processing to text fields.[20]

We identified all adults with a diagnosis of schizophrenia (International Classification of Diseases 10th Revision: F20.x), who had received paliperidone palmitate prescription for at least 365 days and had at least 1 year of recorded treatment from SLAM, continuous or not, prior to the first recorded receipt of paliperidone palmitate. The first recorded receipt of paliperidone palmitate was defined as the 'index date' for definitions

of outcomes and covariates and all patients were followed up for 1 year from their index date. The observation period was from 2011 (when paliperidone palmitate was introduced in the UK) to 2016 (when data were extracted for this analysis).

A range of covariates were identified for cohort characterisation. These included age on the index date, gender and recorded ethnicity grouped into the following categories: white, black, Asian or mixed/other. Marital status was derived from a clinician-completed compulsory structured field in the source record, and was grouped into the following categories: single; married/civil partnership/cohabiting; divorced/separated; widowed. Employment status was derived from compulsory structured fields and referred to the most recent recording on or prior to the index date. Smoking status was derived from a combination of structured fields in the record and a natural language processing algorithm used to ascertain free text inferring to smoking status[17]; current or previous smoking in the record was ascertained prior to the index date. The Health of the Nation Outcome Scales (HoNOS) instrument is routinely used in SLAM services for monitoring purposes, and individual scales from this instrument were extracted as 5-point scores (no problem, minimal, mild, moderate and severe problem). HCP contacts were quantified from structured metadata for SLAM case note entries, ascertaining contacts on the basis of an 'attended' encounter recorded as either face to face or by phone. The profession of the person posting each entry was ascertained from a SLAM Human Resources database of named staff mapped to professional groups. Within the intervals of interest, durations of time were also calculated that each individual spent as an inpatient, and 'active' to SLAM (ie, receiving assessment/care and not discharged). The category of 'medical' refers to trainee psychiatrists.

### Patient and public involvement

The project was reviewed and approved by the CRIS Oversight Committee. All CRIS-related research projects are considered and approved by a patient-led Oversight Committee, reporting to the Caldicott Guardian. The committee considers the appropriateness of the research proposals and adjudicates on risks of deanonymisation at the analysis planning stage.

### Statistical analyses

We used STATA V.13 for all statistical analyses. Sample characteristics were summarised for the total cohort. We compared HCP use between the 12 months before and after initiation, choosing this time period a priori due to the risk of selection (attrition) bias for longer postinitiation periods. Mean numbers of HCP contacts were compared between the two periods of interest within individuals in the sample using Wilcoxon signed-rank tests. The level of significance was p value <0.05. The median and IQR are also provided for all HCP contacts. Figure 1 describes all methodologies adopted to investigate the

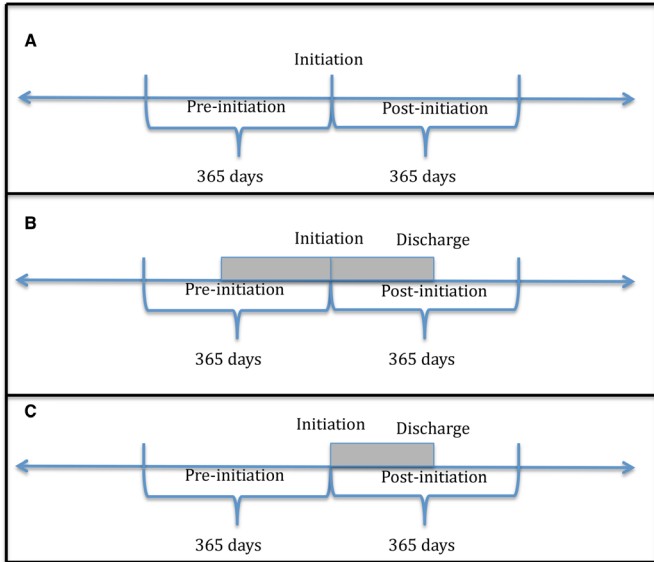

**Figure 1** (A) The unadjusted model took into account all preinitiation and postinitiation time for both inpatients and outpatients. (B) Mirror 1 approach—for inpatients, all time between initiation and inpatient discharge was discarded, as well as an equivalent amount of time preinitiation. (C) Mirror 2 approach—for inpatients, the time between initiation and inpatient discharge was discarded. Shaded area reflects inpatient time that was discarded.

data: (a) The unadjusted analysis reflects the unmodified preinitiation/postinitiation comparisons of HCP contacts. (b) In mirror approach 1, for inpatients, all time between initiation and inpatient discharge was discarded, as well as an equivalent amount of time preinitiation to minimise potential bias from extra inpatient days required for paliperidone palmitate stabilisation, while ensuring that comparisons were made between periods of identical duration. For all outpatients, no time was excluded. (c) In mirror approach 2, for service users who were inpatients at the time of their paliperidone palmitate initiation, the time between paliperidone palmitate initiation and inpatient discharge was excluded, as this is the period when the new medication is stabilising. In the latter mirror image approach, the days between hospitalisation and paliperidone palmitate initiation are attributed to the 'before' period, following the assumption that the hospitalisation period is due to the treatment failure of the previous antipsychotic. For all outpatients, no time was excluded.

## RESULTS

We identified 664 patients with schizophrenia who met the inclusion criteria; 375 (56.5%) had paliperidone palmitate initiated in inpatient settings and 289 (43.5%) had paliperidone palmitate initiated in outpatient settings. Of people with schizophrenia receiving paliperidone palmitate for at least a year, 39.5% (n=262) were documented as receiving it for at least 3 years. Online supplemental table 1 describes the sociodemographic, socioeconomic

and clinical symptoms characteristic of the cohort. The mean age was 43 years, 60% of the sample were male, 60% of black ethnicity, close to 80% were single and over 80% were unemployed. Close to three-quarters were recorded as being current smokers. From the most recent HoNOS scores, significant overactive/agitated behaviour was present in around 30% of the sample at initiation, around 25% had problematic alcohol or substance use, around 20% had significant cognitive problems, over 60% had significant hallucinations and/or delusions; however, the prevalence of depressive and self-harm problems was both below 20%. Significant difficulties in social relationships were present in approximately 40%, impaired activities of daily living in around 30%, significant problems with living conditions in over 20% and impaired occupational/recreational activity in around 35%.

Table 1 summarises the unmodified analyses comparing mental healthcare and HCP use before and after paliperidone palmitate initiation. Overall, face-to-face contacts were significantly lower after initiation for nurses (mean difference −2.38, 95% CI −3.69 to −1.07), social workers (mean difference −0.64, 95% CI −1.08 to −0.19), medical professionals (mean difference −0.58, 95% CI −0.81 to −0.34) and unspecified staff members (mean difference −0.42, 95% CI −0.76 to −0.07). However, contact with clinical psychologists was increased in the postinitiation period (mean difference 0.27, 95% CI −0.01 to 0.56). Contact with occupational therapists in the postinitiation period was also higher but difference did not reach statistical significance. Telephone HCP contacts showed similar differences. Inpatient bed days (mean difference 13.93, 95% CI 7.31 to 20.54) and days 'active' to SLAM (mean difference 40.67, 95% CI 33.39 to 47.95) were significantly higher during the postinitiation period.

Table 2 describes the results for the first mirror image analysis, where paliperidone palmitate postinitiation inpatient period was excluded as well as an equivalent period preinitiation. The postinitiation period was characterised with significantly lower face-to-face contact with social workers (mean difference −0.06, 95% CI −0.49 to 0.37), medical professionals (mean difference −0.31, 95% CI −0.54 to −0.08) and unspecified professionals (mean difference −0.41, 95% CI −0.76 to −0.07). The postinitiation period was also characterised with significantly lower telephone contact with social workers (mean difference −0.42, 95% CI −0.76 to −0.07), medical professionals (mean difference −0.08, 95% CI −0.13 to −0.03) and unspecified professionals (mean difference −0.17, 95% CI −0.29 to −0.06). However, face-to-face and telephone contacts with clinical psychologists (face to face mean difference 0.27 (95% CI −0.01 to 0.56); telephone mean difference 0.05 (95% CI −0.03 to 0.13)) were significantly higher than during the preinitiation period. In the postinitiation period, telephone contact with occupational therapist was also significantly higher (mean difference 0.27, 95% CI 0.05 to 0.50). The postinitiation period was also characterised in this model by fewer inpatient bed days (mean difference −10.48, 95% CI −15.74

**Table 1** Frequencies of healthcare professional contacts during the 12-month period before and after the index date—unmodified analysis of the total time periods

| Type of contact | Mean±SD number of contacts | | Median (IQR) | | |
| --- | --- | --- | --- | --- | --- |
| | Before paliperidone palmitate initiation | After paliperidone palmitate initiation | Before paliperidone palmitate initiation | After paliperidone palmitate initiation | P value* |
| Face-to-face contact | | | | | |
| Total | 29.70±24.90 | 25.8±20.40 | 26 (24) | 22 (17) | <0.001 |
| Nurse | 17.60±16.90 | 15.3±12.74 | 15 (22) | 13 (11) | <0.001 |
| Social worker | 4.00±5.90 | 3.32±6.12 | 2 (5) | 1 (3) | <0.001 |
| Medical | 1.74±3.00 | 1.16±2.13 | 1 (2) | 0 (1) | <0.001 |
| Consultant | 1.76±2.57 | 1.53±2.20 | 1 (2) | 1 (2) | 0.07 |
| Occupational therapist | 0.70±2.51 | 0.94±3.30 | 0 (0) | 0 (0) | 0.41 |
| Administrative | 1.07±2.90 | 0.97±3.10 | 0 (1) | 0 (1) | 0.09 |
| Unspecified | 1.33±4.66 | 0.91±3.11 | 0 (1) | 0 (0) | <0.001 |
| Healthcare assistant | 0.83±6.23 | 0.76±3.69 | 0 (0) | 0 (0) | 0.34 |
| Clinical psychologist | 0.58±2.61 | 0.85±3.63 | 0 (0) | 0 (0) | 0.04 |
| Other therapist | 0.08±0.77 | 0.08±0.96 | 0 (0) | 0 (0) | 0.44 |
| Phone contact | | | | | |
| Total | 11.03±10.45 | 8.74±8.74 | 8 (13.5) | 6 (11) | <0.001 |
| Nurse | 6.06±7.41 | 4.89±5.86 | 3 (8) | 3 (6) | <0.001 |
| Social worker | 2.98±5.73 | 2.01±4.65 | 1 (3) | 0 (2) | <0.001 |
| Medic | 0.21±0.82 | 0.10±0.40 | 0 (0) | 0 (0) | 0.001 |
| Consultant | 0.18±0.74 | 0.13±0.48 | 0 (0) | 0 (0) | 0.87 |
| Occupational therapist | 0.52±2.22 | 0.71±2.80 | 0 (0) | 0 (0) | 0.19 |
| Administrative | 0.39±1.20 | 0.34±1.12 | 0 (0) | 0 (0) | 0.20 |
| Unspecified | 0.39±1.43 | 0.22±0.80 | 0 (0) | 0 (0) | 0.001 |
| Healthcare assistant | 0.10±0.60 | 0.08±0.58 | 0 (0) | 0 (0) | 0.13 |
| Clinical psychologist | 0.19±0.92 | 0.24±1.06 | 0 (0) | 0 (0) | 0.03 |
| Other therapist | 0.02±0.21 | 0.01±0.18 | 0 (0) | 0 (0) | 0.21 |
| Inpatient bed days | 45.06±64.00 | 58.99±87.84 | 20 (61) | 23.5 (79.5) | 0.01 |
| Days active† | 322.81±95.63 | 363.28±13.54 | 365 (0) | 365 (0) | <0.001 |

*Testing the individual differences between the two periods using Wilcoxon signed-rank test.
†Days active are calculated from the total number of days on the SLAM case load during the relevant period—that is, excluding days between SLAM referrals.
SLAM, South London and Maudsley NHS Foundation Trust.

to −5.22) and an increase in days active to SLAM services (mean difference 40.67, 95% CI 33.39 to 47.94).

Table 3 illustrates the analysis for the second mirror image approach, where the period between paliperidone palmitate initiation and inpatient discharge was excluded. Overall face-to-face contacts were lower after paliperidone palmitate initiation—this was statistically significant for nurses (mean difference −2.38, 95% CI −3.69 to −1.07), social workers (mean difference −0.64, 95% CI −1.08 to −0.19), medical professionals (mean difference −0.58, 95% CI −0.81 to −0.34) and unspecified staff members (mean difference −0.42, 95% CI −0.76 to −0.07). In addition, telephone contacts were significantly lower after paliperidone palmitate initiation for nurses (mean difference −1.17, 95% CI −1.73 to −0.60), social workers (mean difference −0.97, 95% CI −1.35 to −0.59), medical staff (mean difference −0.11, 95% CI −0.17 to −0.05) and

unspecified staff members (mean difference −0.17, 95% CI −0.29 to −0.06). We further observed increased face-to-face and telephone contacts with clinical psychologists (mean difference −0.27 (95% CI −0.02 to 0.56) and mean difference 0.05 (95% CI −0.03 to 0.14), respectively). There was also a significant decrease in inpatient bed days postinitiation (mean difference −23.96, 95% CI −30.01 to −17.92) and higher number of days 'active' to SLAM (mean difference 40.69, 95% CI 33.39 to 47.94).

## DISCUSSION

To our knowledge, this is the first study to investigate HCP use in secondary mental healthcare associated with paliperidone palmitate initiation. In a large mental healthcare database, we found that the initiation of paliperidone palmitate was associated with significantly decreased

**Table 2** Frequencies of healthcare professional contacts during the 12-month period before and after the index date—'mirror' analysis excluding the influence of the inpatient period around paliperidone palmitate initiation (mirror approach 1)

| Type of contact | Mean±SD number of contacts | | Median (IQR) | | |
| --- | --- | --- | --- | --- | --- |
| | Before paliperidone palmitate initiation | After paliperidone palmitate initiation | Before paliperidone palmitate initiation | After paliperidone palmitate initiation | P value* |
| Face-to-face contact | | | | | |
| Total | 26.87±24.92 | 25.79±20.37 | 23 (24) | 22 (17) | 0.29 |
| Nurse | 16.25±16.62 | 15.25±12.74 | 13 (21) | 13 (11) | 0.40 |
| Social worker | 3.39±5.58 | 3.33±6.12 | 1 (4) | 1 (3) | 0.03 |
| Medical staff | 1.47±2.86 | 1.16±2.13 | 0 (2) | 0 (1) | <0.01 |
| Consultant | 1.76±2.57 | 1.53±2.20 | 1 (2) | 1 (2) | 0.07 |
| Occupational therapist | 0.62±2.34 | 0.94±3.30 | 0 (0) | 0 (0) | 0.09 |
| Administrative | 0.97±2.81 | 0.97±3.09 | 0 (1) | 0 (1) | 0.59 |
| Unspecified | 1.33±4.66 | 0.91±3.11 | 0 (1) | 0 (0) | <0.001 |
| Healthcare assistant | 0.77±6.14 | 0.76±3.69 | 0 (0) | 0 (0) | 0.07 |
| Clinical psychologist | 0.58±2.61 | 0.85±3.63 | 0 (0) | 0 (0) | 0.04 |
| Other therapist | 0.07±0.77 | 0.08±0.96 | 0 (0) | 0 (0) | 0.44 |
| Phone contact | | | | | |
| Total | 9.38±9.57 | 8.74±8.74 | 6.5 (12) | 6 (11) | 0.03 |
| Nurse | 5.28±6.71 | 4.89±5.86 | 3 (8) | 3 (6) | 0.29 |
| Social worker | 2.43±5.18 | 2.01±4.65 | 1 (2) | 0 (2) | <0.001 |
| Medical staff | 0.18±0.75 | 0.10±0.40 | 0 (0) | 0 (0) | 0.02 |
| Consultant | 0.18±0.74 | 0.13±0.48 | 0 (0) | 0 (0) | 0.87 |
| Occupational therapist | 0.43±1.99 | 0.71±2.80 | 0 (0) | 0 (0) | 0.01 |
| Administrative | 0.34±1.17 | 0.34±1.12 | 0 (0) | 0 (0) | 0.98 |
| Unspecified | 0.39±1.43 | 0.22±0.80 | 0 (0) | 0 (0) | <0.01 |
| Healthcare assistant | 0.08±0.55 | 0.08±0.58 | 0 (0) | 0 (0) | 0.33 |
| Clinical psychologist | 0.19±0.92 | 0.24±1.06 | 0 (0) | 0 (0) | 0.03 |
| Other therapist | 0.02±0.21 | 0.01±0.18 | 0 (0) | 0 (0) | 0.21 |
| Inpatient bed days | 31.58±51.99 | 21.09±52.93 | 0 (47) | 0 (11.5) | <0.001 |
| Days active† | 322.81±95.63 | 363.48±12.53 | 365 (0) | 365 (0) | <0.001 |

*Testing the individual differences between the two periods using Wilcoxon signed-rank test.
†Days active are calculated from the total number of days on the SLAM case load during the relevant period—that is, excluding days between SLAM referrals.
SLAM, South London and Maudsley NHS Foundation Trust.

face-to-face and telephone contacts with medical professionals, social workers and unspecified HCP, and increased contact with therapeutic services such as clinical psychologists. This, in addition to the higher proportion of time spent 'active' in SLAM after initiation and a decrease in days spent as an inpatient, is consistent with a shift from treatment and risk management to rehabilitation. The results were largely consistent irrespective of the mirror image approach that was employed to take into account the potentially biasing effects of inpatient care at the time of treatment initiation, although reduction in nursing input postinitiation was only evident in the unadjusted analysis and second mirror image approach (but not in mirror image approach 1).

Considering hospitalisation, our results are consistent with previous research findings that paliperidone palmitate initiation is associated with a reduction in inpatient bed days after the two mirror approaches were applied[18]; we were able to replicate this in a larger cohort of patients initiated on paliperidone palmitate than previously reported, as well as using within-individual comparisons that will have removed the influence of between-individual differences as confounding factors. Taylor and Olofinjana have argued that there are several possible mechanisms that may explain this assumed effect of paliperidone palmitate, including the ability to give loading doses and higher equivalent doses compared with risperidone depot. However, research comparing paliperidone palmitate to other long-acting injectables such as aripiprazole and haloperidol has reported that paliperidone palmitate is prescribed at a lower dose and after 12 months has a similar reduction in urgent consultation, psychiatric hospitalisations and psychiatric symptoms.[22] Ultimately, a formal randomised controlled trial would be required to assess this.

**Table 3** Frequencies of service staff contacts during the 12-month period before and after the index date—'mirror' analysis excluding the influence of the time between paliperidone palmitate initiation and inpatient discharge (mirror approach 2)

| Type of contact | Mean±SD | | Median (IQR) | | P value* |
|---|---|---|---|---|---|
| | Before paliperidone palmitate initiation | After paliperidone palmitate initiation | Before paliperidone palmitate initiation | After paliperidone palmitate initiation | |
| Face-to-face contact | | | | | |
| Total | 29.68±24.85 | 25.79±20.37 | 26 (24) | 22 (17) | <0.001 |
| Nurse | 17.64±16.89 | 15.25±12.74 | 15 (22) | 13 (11) | <0.001 |
| Social worker | 3.96±5.89 | 3.33±6.12 | 2 (5) | 1 (3) | <0.001 |
| Medical staff | 1.74±2.97 | 1.16±2.13 | 1 (2) | 0 (1) | <0.001 |
| Consultant | 1.76±2.57 | 1.53±2.20 | 1 (2) | 1 (2) | 0.07 |
| Occupational therapist | 0.70±2.51 | 0.94±3.30 | 0 (0) | 0 (0) | 0.41 |
| Administration | 1.07±2.90 | 0.97±3.10 | 0 (1) | 0 (1) | 0.09 |
| Unspecified | 1.33±4.66 | 0.91±3.11 | 0 (1) | 0 (0) | <0.001 |
| HCA | 0.83±6.23 | 0.76±3.69 | 0 (0) | 0 (0) | 0.34 |
| Clinical psychologist | 0.58±2.61 | 0.85±3.63 | 0 (0) | 0 (0) | 0.04 |
| Other therapist | 0.08±0.77 | 0.08±0.96 | 0 (0) | 0 (0) | 0.44 |
| Phone contact | | | | | |
| Total | 11.03±10.45 | 8.74±8.74 | 8 (13.5) | 6 (11) | <0.001 |
| Nurse | 6.06±7.41 | 4.89±5.86 | 3 (8) | 3 (6) | <0.001 |
| Social worker | 2.98±5.73 | 2.01±4.65 | 1 (3) | 0 (2) | <0.001 |
| Medical staff | 0.21±0.82 | 0.10±0.40 | 0 (0) | 0 (0) | <0.001 |
| Consultant | 0.18±0.74 | 0.13±0.48 | 0 (0) | 0 (0) | 0.87 |
| Occupational therapist | 0.52±2.22 | 0.71±2.80 | 0 (0) | 0 (0) | 0.19 |
| Administration | 0.39±1.20 | 0.34±1.12 | 0 (0) | 0 (0) | 0.20 |
| Unspecified | 0.39±1.43 | 0.22±0.80 | 0 (0) | 0 (0) | 0.001 |
| HCA | 0.10±0.60 | 0.08±0.58 | 0 (0) | 0 (0) | 0.13 |
| Clinical psychologist | 0.19±0.92 | 0.24±1.06 | 0 (0) | 0 (0) | 0.03 |
| Other therapist | 0.02±0.21 | 0.01±0.18 | 0 (0) | 0 (0) | 0.21 |
| Inpatient bed days | 45.06±64.00 | 21.09±52.93 | 20 (61) | 0 (11.5) | <0.001 |
| Days active† | 322.81±95.63 | 363.48±12.53 | 365 (0) | 365 (0) | <0.001 |

*Testing individual differences between the two periods using Wilcoxon signed-rank test.
†Days active are calculated from the total number of days on the SLAM case load during the relevant period—that is, excluding days between SLAM referral.
HCA, healthcare assistant; SLAM, South London and Maudsley NHS Foundation Trust.

The postinitiation reduction in contact with medical and social work staff in both models, and reduction in nursing staff contact in the second model, could reflect improvements in mental state and is consistent with a wider benefit of treatment initiation than is indicated by inpatient care comparisons. Furthermore, the increased contact with clinical psychology staff does not support an overall reduction in HCP contact but instead an altered profile of contact from staff members responsible for treatment compared with those responsible for rehabilitation. It should be borne in mind that no control condition was included in the design—that is, the apparent HCP changes might have been observed regardless of the date chosen to divide comparison periods, and/or samples receiving inpatient care may naturally have higher medical and social work involvement prior to an admission episode. Furthermore, if changes in HCP contacts represent an effect of the intervention rather than inpatient care, it was not possible to subdivide the sample in terms of conditions prior to the event (ie, the precise medication change being instituted). Further research could benefit from examining this in the future. Findings from previous observational research have indicated that as compared with risperidone long-acting injectables, patients prescribed paliperidone palmitate are more likely to be adherent to their treatment, to have reduced hospitalisation and fewer emergency department visits.[23] Furthermore, in France and Sweden, paliperidone palmitate has been found to be more cost-effective and to have a lower cost over 5 years, as compared with other antipsychotics such as risperidone, aripiprazole and haloperidol.[24 25]

This study had several strengths. We examined a large and diverse sample of patients in naturalistic settings.

Therefore, our sample should be maximally reflective of HCP contact in real-world clinical settings. The use of mental healthcare data was justified because very few paliperidone palmitate initiations would be expected outside specialist services. Furthermore, we had sufficient statistical power to characterise the sample according to a diverse number of patient characteristics, and the within-patient comparisons remove the effect of between-patient differences as confounding factors. Finally, in addition to the unadjusted analysis, we employed two methods of mirror image analyses to address the potentially biasing impact of the inpatient episodes within which many initiation events took place.

On the other hand, several limitations need to be borne in mind. The study was observational in nature and therefore the before-after comparisons cannot be assumed to represent causal relationships (ie, a treatment effect). In addition, the data were limited by availability from a routine health record and therefore we were unable to examine the factors underlying changes in HCP contacts, such as clinical symptoms before and after paliperidone palmitate and medication history. In addition, selecting patients who have taken paliperidone palmitate for a year is likely to represent a group of patients who have good outcomes. Mace and colleagues[26] have indicated that patients with longer duration of illness, who have been initiated on long-acting injectables such as haloperidol decanoate, for reasons other than non-adherence, are more likely to remain on their treatment. Furthermore, we did not examine the formulation of paliperidone palmitate (eg, monthly; three monthly); however, at the time the study was conducted we believe the monthly formulation was mostly in use. In addition, a longer follow-up period would have allowed us to investigate the longer term effect of paliperidone palmitate and therefore should be considered by further research. It is also important to consider that follow-up began at the point paliperidone palmitate was initiated as opposed to hospital discharge. Furthermore, we did not include a control group of another depot or oral antipsychotic, therefore we cannot infer that changes associated with paliperidone palmitate initiation would not have been observed with other interventions. Lastly, we did not make adjustments for multiple comparisons in the analyses.

Overall, our findings suggest that paliperidone palmitate initiation is associated with a change in HCP contact which could indicate change in patients' presentation. However, further research is needed that focuses on examining changes in mental health symptoms from the before period to after period to support this hypothesis. Finally, we did not seek to quantify the economic impact of changes in health service use, and further research would be needed to determine whether this translates into lower overall service costs.

**Contributors** GK-S drafted the manuscript and together with DA conducted the analyses. MB, MP and HS put together the data extraction plan and extracted the data. AB, C-KC, DT, RH and RS contributed to the conceptualisation and designing of the study. All authors critically revised the manuscript and have approved the final version. GK-S, RH and RS are acting as guarantors.

**Funding** The study described here was funded by Janssen UK. This work was supported by the Clinical Record Interactive Search (CRIS) platform funded and developed by the National Institute for Health Research (NIHR) Biomedical Research Centre at South London and Maudsley NHS Foundation Trust and King's College London, and a joint infrastructure grant from Guy's and St Thomas' Charity and the Maudsley Charity (grant number BRC-2011-10035).

**Disclaimer** The views expressed are those of the authors and not necessarily those of the NHS, the NIHR or the Department of Health.

**Competing interests** RH, C-KC, HS and RS have received research funding from Roche, Pfizer, Janssen and Lundbeck. GK-S has received funding from Janssen and Lundbeck. RS has also received research funding from Takeda. AB is an employee of Janssen UK. DT has received investigator-initiated research grants from AstraZeneca, Eli Lilly, Janssen, Lundbeck, Otsuka, Servier and Sunovion. RH, RS, HS, C-KC and GK-S receive salary support from the National Institute for Health Research (NIHR) Biomedical Research Centre at South London and Maudsley NHS Foundation Trust and King's College London. DT is a pharmacist and clinical academic at the South London and Maudsley Hospital NHS Trust and King's College London.

**Patient and public involvement** Patients and/or the public were involved in the design, or conduct, or reporting, or dissemination plans of this research. Refer to the Methods section for further details.

**Patient consent for publication** Not required.

**Ethics approval** CRIS was approved for secondary analysis by the Oxfordshire Research Ethics Committee C (reference 18/SC/0372).

**Provenance and peer review** Not commissioned; externally peer reviewed.

**Data availability statement** Data are available upon reasonable request. The data that support the findings of this study are available on request from the corresponding author. The data are not publicly available due to the information governance framework and REC approval in place concerning CRIS data use.

**ORCID iD**
Giouliana Kadra-Scalzo http://orcid.org/0000-0003-3182-905X

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
