## [Reviewer comments · BMJ Open]

ARTICLE DETAILS

TITLE (PROVISIONAL)	Mental health care utilisation by patients before and after receiving paliperidone palmitate treatment: mirror-image analyses.
AUTHORS	Kadra-Scalzo, Giouliana; Ahn, Deborah; Bird, Alex; Broadbent, Matthew; Chang, Chin-Kuo; Pritchard, Megan; Shetty, Hitesh; Taylor, David; Hayes, Richard; Stewart, Robert

VERSION 1 – REVIEW

REVIEWER	Deslandes, Paul University Hospital Llandough, All Wales Therapeutics and Toxicology Centre, Academic Centre
REVIEW RETURNED	13-Oct-2021

GENERAL COMMENTS	This is an interesting manuscript, which investigates the outcomes of patients treated with the antipsychotic paliperidone palmitate in clinical practice. Such studies help to supplement RCT data, through the inclusion of a patient cohort that is representative of those treated in routine practice. I have some comments relating to aspects of the manuscript as follows: Background, Page 5, paragraph 2, lines 31 to 40. It would be useful for the reader if the naturalistic, observational nature of the evidence presented in this paragraph were highlighted, and the associated limitations noted. Background, Page 6, paragraph 1, lines 15 to 22: The sentence starting “However, although hospitalisation episodes...” would benefit from the addition of a supporting reference. Background, Page 6, lines 31 to 35: The stated aim of the study included “...comparing the frequency and duration of HCP contacts...”. Whilst the study reports the duration of hospitalisation prior to and following PP initiation, and time active to SLAM, it does not report the duration of any HCP contacts. Although this is not explicitly stated as an aim of the study, the description of the aim might be a little misleading in this regard. It may be clearer to explain that “duration” related to inpatient stay, and time active, and that number of HCP contacts but not their duration was captured. Methods: I could not see any mention of the time period over which the study was conducted, and when the last index date was. If the final index date was in 2019/20, could the Covid-19 pandemic have had an impact on HCP contact?
---

	Methods, Page 8, paragraph 1, first line: meaning a little unclear/typo. Methods, Page 9, lines 21 to 41: I found the explanation of the excluded time periods for inpatients in mirror 1 and mirror 2 a little unclear. Was the same approach taken in both analyses? i.e. the time from PP initiation to inpatient discharge was excluded in both cases. If so, it would be clearer for the reader if the same description was used in each case. Currently, mirror 1 is explained as "...the post-initiation inpatient period was excluded...", whilst mirror 2 is explained as "...the period between PP initiation and inpatient-discharge was excluded...". Having subsequently read the legend of Figure 1, I believe the same approach was taken, and I would suggest making the text in the methods section consistent with that of the legend of figure 1. Methods, Statistical analysis Pages 8/9: The assumed level of significance for statistical analysis is not stated here, nor whether adjustment was made for multiple comparisons. Results, Page 10, paragraph 1, line 9: Is "%" missing from the figures in parentheses? Results, Tables 3 and 4: Given that the data for the post intervention period are the same for mirror 1 and mirror 2 analyses, tables 3 and 4 could be combined to avoid having to duplicate that same set of data in both tables. Discussion generally: The discussion does not include many cited references, and thus does not always place this work in the context of other literature. Whilst the authors note that to their knowledge this is the first study to investigate HCP usage associated with PP, there were perhaps some missed opportunities to consider the implications of this work in the wider context. For example, there are some cost-effectiveness analyses of PP in the published literature, which assess healthcare utilisation albeit in different settings and using different methodologies. Discussion/Results/Tables 2-4: Although statistical differences were observed in relation to certain HCP contacts, the numerical differences were often small (e.g. Table 4 social worker mean pre PP 3.96 vs post PP 3.33). Could the authors comment on the relevance of these small differences (for example from a health economic perspective, or from a clinical perspective), and perhaps cite some pertinent literature to help to address the comment above. Discussion/Results/Tables 2-4: It appears that medical contact does not include consultant contact, as these are presented separately in the tables. Assuming that consultant contact relates to a medical consultant, it would be helpful if the authors could explain the rationale for treating these separately. It would also be interesting to know whether statistical significance remains if medical and consultant contact are combined and analysed as a single entity. Discussion, page 13, paragraph 1, lines 11-25, and page 14 paragraph 2 lines 9-23: These sections make the assumption that nurse, social work and medical contact is primarily about risk
--	---

	management, and that clinical psychology is about rehabilitation. Can you cite literature to support this, particularly in relation to social work contact, which might involve a variety of tasks? As noted later in the paragraph, it would be interesting to know whether other factors (such as an inpatient admission) were a driver for the assumed change from risk management to rehabilitation. The authors state that "...it was not possible to subdivide the sample in terms of conditions prior to the event...", but 375 patients were initiated as inpatients, and 289 as outpatients, which seems like it could form the basis for a comparison of inpatient vs outpatient status at initiation as a confounder. Discussion, page 14, paragraph 2, line 13/14: "...may reflect similar improvements..." similar to what, you only have one cohort of patients?
--	--

REVIEWER	Ceskova, Eva University Hospital Ostrava, Department of Psychiatry
REVIEW RETURNED	13-Dec-2021

GENERAL COMMENTS	The topic is very actual. The cost effectiveness of second generation long- acting antipsychotic is under discussion. The authors used besides classical outcome measures some other outcomes which may be important for long-term prognosis. Following initiation of paliperidone palmitate the inpatient bed days were lower and face -to -face and telephone contacts with medical and social work health professional were lower, however higher contact with clinical psychologists Abstract + conclusion The authors conclude that the change in the profile of health care use is consistent with a transition from treatment to rehabilitation. Suggestion To moderate this statement. Background Suggestion To mention that there is any head- to- head comparison of second generation long-acting antipsychotics concerning to the use of above- mentioned secondary outcomes measures Method Suggestion To add explanation why paliperidone palmitate was chosen Discussion Suggestion In discussion to mention the change of profile use could be early sign of deterioration – i.e., the effort for more autonomy, may be withdrawal of the treatment, lack of insight. It should be taken into consideration, that the 12months period is too short. Further, no symptoms were measured
---

VERSION 1 – AUTHOR RESPONSE

Reviewer: 1

Background, Page 5, paragraph 2, lines 31 to 40. It would be useful for the reader if the naturalistic, observational nature of the evidence presented in this paragraph were highlighted, and the associated limitations noted.

Authors' response:

We have amended the text in this section as follows:

Consequently, the aim of this study was to describe inpatient and community mental healthcare service use, using observational data including HCP contacts, of patients with schizophrenia being treated with paliperidone palmitate, specifically comparing the frequency and duration of HCP contacts before and after initiation of paliperidone palmitate therapy, using mirror-image methodologies. On one hand, observational data are limited with respect to trial data because of the lack of randomisation and thus the risk of residual confounding; on the other hand, they provide an opportunity to assess associations in a more naturalistic and generalisable setting than is generally the case for trials.

We have thus incorporated what we feel are the key methodological considerations, although believe that a full discussion of strengths and limitations is best located in the Discussion section which includes a further consideration of the limitations of this research at length (page 14).

Background, Page 6, paragraph 1, lines 15 to 22: The sentence starting "However, although hospitalisation episodes..." would benefit from the addition of a supporting reference.

Authors' response:

We have added a reference to this sentence.

Background, Page 6, lines 31 to 35: The stated aim of the study included "...comparing the frequency and duration of HCP contacts...". Whilst the study reports the duration of hospitalisation prior to and following PP initiation, and time active to SLAM, it does not report the duration of any HCP contacts. Although this is not explicitly stated as an aim of the study, the description of the aim might be a little misleading in this regard. It may be clearer to explain that "duration" related to inpatient stay, and time active, and that number of HCP contacts but not their duration was captured.

Authors' response:

We have edited the text in question to make these changes (see quoted text in the response above)

Methods: I could not see any mention of the time period over which the study was conducted, and when the last index date was. If the final index date was in 2019/20, could the Covid-19 pandemic have had an impact on HCP contact?

Authors' response:

We have added text to this section to confirm that the observation period was from 2011 (when paliperidone palmitate was introduced in the UK) to 2016 (when data were extracted for this analysis); therefore there was no impact of the COVID-19 pandemic. (see page 6)

Methods, Page 8, paragraph 1, first line: meaning a little unclear/typo.

Authors' response:

This has been changed to: Smoking status was derived from a combination of structured fields in the record and a natural language processing algorithm used to ascertain free-text inferring to smoking status

Methods, Page 9, lines 21 to 41: I found the explanation of the excluded time periods for inpatients in mirror 1 and mirror 2 a little unclear. Was the same approach taken in both analyses? i.e. the time from PP initiation to inpatient discharge was excluded in both cases. If so, it would be clearer for the reader if the same description was used in each case. Currently, mirror 1 is explained as "...the post-initiation inpatient period was excluded...", whilst mirror 2 is explained as "...the period between PP initiation and in-patient-discharge was excluded...". Having subsequently read the legend of Figure 1, I believe the same approach was taken, and I would suggest making the text in the methods section consistent with that of the legend of figure 1.

Authors' response:

This section has been reworded to make it more consistent with the legend of Figure 1:

b) in mirror approach 1, for inpatients all time between initiation and inpatient discharge was discarded, as well as an equivalent amount of time pre-initiation to minimise potential bias from extra inpatient days required for paliperidone palmitate stabilisation, whilst ensuring that comparisons were made between periods of identical duration. For all outpatients no time was excluded; c) in mirror approach 2, for service users who were inpatients at the time of their paliperidone palmitate initiation, the time between paliperidone palmitate initiation and inpatient discharge was excluded, as this is the period when the new medication is stabilising. In the latter mirror image approach, the days between hospitalisation and paliperidone palmitate initiation are attributed to the 'before' period, following the assumption that the hospitalisation period is due to the treatment failure of the previous antipsychotic (3). For all outpatients no time was excluded.

Methods, Statistical analysis Pages 8/9: The assumed level of significance for statistical analysis is not stated here, nor whether adjustment was made for multiple comparisons.

Authors' response:

The level of significance was a p value less than 0.05 (this has been added to the methods section on page 8). We did not make adjustments for multiple comparisons and have added this to the limitations section (page 14).

Results, Page 10, paragraph 1, line 9: Is "%" missing from the figures in parentheses?

Authors' response:

% has been added.

Results, Tables 3 and 4: Given that the data for the post intervention period are the same for mirror 1 and mirror 2 analyses, tables 3 and 4 could be combined to avoid having to duplicate that same set of data in both tables.

Authors' response:

Although mirror 1 and mirror 2 look very similar, the time that has been discarded for both approaches is different. In mirror approach 1, for inpatients all time between initiation and inpatient discharge was discarded, as well as an equivalent amount of time pre-initiation, whereas in mirror approach 2, for inpatients at the time of their paliperidone palmitate initiation, the time between paliperidone palmitate initiation and in-patient discharge was excluded. We therefore feel that the tables should be kept separated to demonstrate the equivalence, as this would be difficult to justify with a simple text statement; however, we are happy to accept an Editorial judgement on the matter.

Discussion generally: The discussion does not include many cited references, and thus does not always place this work in the context of other literature. Whilst the authors note that to their knowledge this is the first study to investigate HCP usage associated with PP, there were perhaps some missed opportunities to consider the implications of this work in the wider context. For example, there are some cost-effectiveness analyses of PP in the published literature, which assess healthcare utilisation albeit in different settings and using different methodologies.

Authors' response:

We have now added literature and discussion around the wider context of PP prescription (see highlighted paragraphs on pages 12 and 13).

Discussion/Results/Tables 2-4: Although statistical differences were observed in relation to certain HCP contacts, the numerical differences were often small (e.g. Table 4 social worker mean pre PP 3.96 vs post PP 3.33). Could the authors comment on the relevance of these small differences (for example from a health economic perspective, or from a clinical perspective), and perhaps cite some pertinent literature to help to address the comment above.

Authors' response:

The clinical differences are relatively small; however, when multiplied over the large numbers of patients requiring long-acting antipsychotics and the duration of time they're receiving them, these may nevertheless translate into benefits at a group level. We have also added in the Discussion section (page 12 and 13), as suggested above, other observational research, some of which has also reported similarly small but significant changes.

Discussion/Results/Tables 2-4: It appears that medical contact does not include consultant contact, as these are presented separately in the tables. Assuming that consultant contact relates to a medical consultant, it would be helpful if the authors could explain the rationale for treating these separately. It would also be interesting to know whether statistical significance remains if medical and consultant contact are combined and analysed as a single entity.

Authors' response:

These are recorded separately in the electronic health records and represent quite different types of staff contact and reasons for staff contact, as well as potential unit costs. We therefore felt that it would not be appropriate to combine the two, in the same way that we did not combine other professional groups. However, we have added text to clarify this in the Methods on page 7 (The category of 'medical' refers to trainee psychiatrists.)

Discussion, page 13, paragraph 1, lines 11-25, and page 14 paragraph 2 lines 9-23: These sections make the assumption that nurse, social work and medical contact is primarily about risk management, and that clinical psychology is about rehabilitation. Can you cite literature to support this, particularly in relation to social work contact, which might involve a variety of tasks? As noted later in the paragraph, it would be interesting to know whether other factors (such as an inpatient admission) were a driver for the assumed change from risk management to rehabilitation. The authors state that "...it was not possible to subdivide the sample in terms of conditions prior to the event...", but 375 patients were initiated as inpatients, and 289 as outpatients, which seems like it could form the basis for a comparison of inpatient vs outpatient status at initiation as a confounder.

Authors' response:

In the UK, social workers are often involved in involuntary hospitalisations, whereas a referral to psychological services is made when the service user is sufficiently stable to engage in therapeutic work. Based on this we made the above conclusions; however, we accept that these are assumptions and have amended the text so that this is made clearer. In relation to the other point made, we feel that it is beyond the scope of the paper and over-exploratory to stratify output further; however, we have added text to acknowledge that this would be a fruitful topic for further research.

Discussion, page 14, paragraph 2, line 13/14: "...may reflect similar improvements..." similar to what, you only have one cohort of patients?

Authors' response:

'Similar' has been removed.

Reviewer: 2

The authors conclude that the change in the profile of health care use is consistent with a transition from treatment to rehabilitation.

Suggestion

To moderate this statement.

Authors' response:

This has been changed to the following text: Our findings indicate a change in the profile of HCP use, consistent with a transition from treatment to possibly a rehabilitation.

We have also added some additional relevant literature (please see sections that have been highlighted).

Background

Suggestion

To mention that there is any head- to- head comparison of second generation long-acting antipsychotics concerning to the use of above- mentioned secondary outcomes measures

Authors' response:

We have added relevant studies and discussed these within the context of our findings on pages 13 and 14.

Method
Suggestion
To add explanation why paliperidone palmitate was chosen

Authors' response:

We now describe this in detail on page 5 of the Background section.

Discussion
Suggestion
In discussion to mention the change of profile use could be early sign of deterioration – i.e., the effort for more autonomy, may be withdrawal of the treatment, lack of insight. It should be taken into consideration, that the 12months period is too short. Further, no symptoms were measured

Authors' response:

We have now highlighted these issues in the limitations section on page 14.

VERSION 2 – REVIEW

REVIEWER	Deslandes, Paul University Hospital Llandough, All Wales Therapeutics and Toxicology Centre, Academic Centre
REVIEW RETURNED	18-Jan-2022

GENERAL COMMENTS	Thank you for addressing my previous comments, I am happy to recommend that the manuscript is accepted for publication.
---

REVIEWER	Ceskova, Eva University Hospital Ostrava, Department of Psychiatry
REVIEW RETURNED	18-Jan-2022

GENERAL COMMENTS	The revision is OK, I have no more suggestions
--